# Research on Frequency Control of Islanded Microgrid with Multiple Distributed Power Sources

**Yirong Liu** [1], **Xiaoli Wang** [1,*] and **Shilin Wang** [2]

1   School of Mechanical, Electrical and Information Engineering, Shandong University, Weihai 264209, China; lyr@mail.sdu.edu.cn
2   HIT(Anshan)Institute of Industrial Technology, Anshan 114000, China; shilinwang1992@163.com
*   Correspondence: wxl@sdu.edu.cn; Tel.: +86-138-6302-6640

**Abstract:** At present, some achievements have been made in the research on the energy management of microgrid operation. However, the research is mainly on the operation of grid-connected microgrid, while the research on the energy management of islanded microgrid is still relatively few. Frequency is one of the characteristics that affects the reliability and power quality of the microgrid. The essence of controlling frequency stability is to maintain source-load balance and redistribution of active power. Therefore, this paper proposes a frequency control strategy based on dynamically cutting machine to reduce load by analyzing the use priority of different distributed power supply and the division of load importance degree, and combining the influence degree of different frequency variation range on microgrid. To coordinate and control distributed power supply, energy storage device, and load in different frequency change areas, this paper proposes different control strategies. The seed strategies of the control strategy are discussed one by one. Experimental results show that the frequency control strategy can significantly improve the frequency stability of the power supply system and reduce the operating cost of islanded microgrid.

**Keywords:** microgrid; zone control; frequency control; islanded microgrid

## 1. Introduction

Historically, fossil fuels such as oil, coal, and natural gas have been the main sources of energy demand growth in the world. However, due to the foreseeable depletion of traditional fossil energy, the world has paid attention to the sustainable development of society. In addition, the large-scale use of traditional fossil energy has also caused serious environmental pollution problems [1]. As an intelligent, environment friendly, and flexible distributed power generation solution, microgrid can effectively solve the contradiction between traditional large power grid and social development and improve the utilization rate of renewable energy. The isolated microgrid, which is not connected to the external large power grid, can provide high-quality power supply to areas where it cannot be covered by the large power grid, such as the marginal mountains and islands [2–5]. However, the unpredictability of load and random output of existing distributed power sources, such as wind power generation and photovoltaic power generation, will have a great impact on the power quality of microgrid [6–9]. If the proportion of distributed power in the large power grid is too high, there will also have a certain disturbance to the large power grid, affecting the safe operation of the large power grid [10,11].

Frequency is an important characteristic that affects the power supply reliability and power quality of microgrid. In recent years, the research on frequency control of microgrid has attracted widespread attention and achieved some achievements. Literature [12] studies showed that the errors in active power sharing caused by differences in instantaneous frequency deviations between microgrid units, and proposes a secondary control method for distributed generation (DG) units, which

is used to implement active power sharing and frequency recovery simultaneously in an islanded microgrid. However, secondary control requires additional communication systems and forced reactive power sharing may result in poor system voltage distribution. In view of the impact of solar power generation on the power quality of microgrid, [13] proposes a decentralized power management and load sharing method for photovoltaic islanded microgrid. A decentralized hierarchical control method was proposed in the literature [14], which improved the power quality, but did not give a specific frequency control strategy. At present, most schemes fail to consider the diversity of distributed power supply and the hierarchy of system frequency control. In fact, according to the frequency characteristic curve of load, when the active power input of microgrid system increases, the system frequency increases. When the active power input of the microgrid system drops, the system frequency decreases [15]. By deploying uninterruptible power within the microgrid, such as a certain capacity of battery energy storage equipment and diesel generating sets, can balance the microgrid system input active power, so as to balance the system frequency [16].

Therefore, for the problems such as small installed capacity of isolated microgrid system, unstable output of distributed power supply, high permeability of renewable energy, and rapid diffusion of abnormal problems. In this paper, a new frequency division control structure of microgrid is designed by using the idea of zonal control and a hierarchical coordinated control strategy is proposed. The frequency is divided into several zones. Different zones stabilize the frequency by adopting different control strategies, cutting load according to priority, distributing power and so on. At the same time, an uninterruptible power supply composed of lead-acid energy storage system and diesel generator set is configured to participate in frequency control. Finally, the stability of the proposed method was verified by a stable operation test in the microgrid laboratory of Shandong University (Weihai, China).

The rest of this paper is organized as follows. Section 2 starts with a brief review of islanded microgrid with multiple distributed power sources, and then the proposed method is described. Frequency stability control strategy of island AC microgrid is designed in Section 3. Experimental results of the proposed method are presented in Section 4. Conclusions and remarks on possible further work are finally given in Section 5.

## 2. Isolated Microgrid with Wind, Light, Diesel Power Generation, and Storage

The purpose of this paper is to provide high-quality power supply for the areas such as remote mountainous areas and small islands that are not covered by traditional large power grids. In this paper, while making full use of solar energy, wind energy, and other pollution-free renewable energy generation, considering that wind power generation or photovoltaic power generation will lose its power generation effect under the weather conditions such as no wind for a long time or no light on a continuous cloudy day, diesel generator set is added and lead-acid battery is used as an energy storage device to form islanded AC microgrid.

### 2.1. Microgrid Topology Introduction

The islanded microgrid consists of 3 kW photovoltaic power generation, 3 kW cage asynchronous wind turbine, 3 kW diesel genset, and 16 kW × 6 h lead-acid battery energy storage system and other protection measurement and control equipment. Through the corresponding control strategy and the islanded microgrid central stability controller, the islanded microgrid can provide a stable, reliable, and high-quality power supply for the load. Its topology diagram is shown in Figure 1.

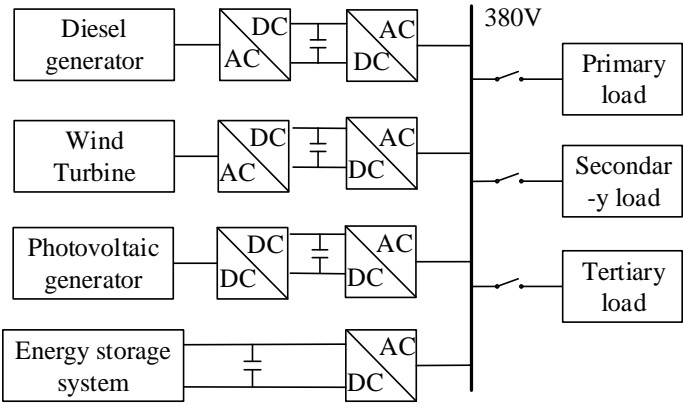

**Figure 1.** The topology of islanded microgrid.

*2.2. Distributed Power and Load Priority Classification*

The islanded microgrid designed in this paper adopts the AC bus structure. The grid is equipped with photovoltaic (PV) inverters, wind turbines (WT), simulated diesel generators, and battery power conversion systems (PCS). The load is prioritized according to different importance levels. The specific classification strategy is shown in Table 1.

**Table 1.** Load prioritization.

| Priority | Load Nature | Description |
|---|---|---|
| Primary load | Sensitive load | Important loads, such as military medical facilities, give priority to ensuring stable operation |
| Secondary load | Non-sensitive load | Non-critical loads, such as residential electricity, can be removed at critical moments |
| Tertiary load | Non-sensitive load | Non-critical loads, such as industrial production, electricity, and other facilities, can be removed at critical moments |

In this paper, the order of use of DG is also prioritized. The specific evaluation criteria are shown in Table 2.

**Table 2.** Distributed power prioritization.

| DG | Diurnal | Night | Description |
|---|---|---|---|
| PV | 1 | 3 | Power generation is stable during the sunshine period, affected by illumination, stable energy, no impact |
| WT | 2 | 1 | Power generation during windy periods, unstable power generation, easy to impact the system |
| Diesel genset | 3 | 2 | Expensive secondary energy generation, uneconomical, stable and reliable, not affected by weather factors |

As shown in Table 2, the priority of DG input mainly considers its stability and economic performance. As the most stable generator, diesel engine is not put into the power generation plan for economic reasons when the daily scenery energy storage meets the daily electricity demand of micro grid. When the daily WT, PV and PCS provides a partial gap or the system has a frequency offset and other faults that require electrical energy input, the diesel gensets is put into use.

## 3. Frequency Stability Control Strategy of Islanded AC Microgrid

Compared with traditional large power grid, microgrid have large differences in terms of structure, power supply, and load. This makes the traditional large power grid control method no longer adapt to the control needs of the microgrid. For this characteristic, the islanded microgrid frequency control

strategy designed in this paper uses $v - f$ control to achieve energy balance in the microgrid. That is, the sum of the generated energy of each distributed generator is equal to the sum of the load power, and the stability of the internal frequency of the microgrid is realized at the same time, as shown in Equations (1) and (2):

$$f_{\min} \leq f(t) \leq f_{\max}, \tag{1}$$

$$\sum P_{DG}(t) = \sum P_{load}(t), \tag{2}$$

where $f(t)$ is the frequency of the current time bus, which should be between the specified upper limit $f_{\max}$ and lower limit $f_{\min}$. Considering the small capacity of the microgrid, the design frequency index of this paper is $f_{\max} = 50.5\text{Hz}$ and $f_{\min} = 49.5\text{Hz}$. Where $\sum P_{DG}(t)$ is the sum of the distributed power of the current microgrid. Distributed power sources include diesel gensets, energy storage systems, photovoltaic generator sets, and wind turbines. The specific expression is shown in Equation (3):

$$\sum P_{DG} = P_{bat}(t) + P_{wind}(t)k_{MTTP\_wind} + P_{solar}(t)k_{MTTP\_solar} + P_{engine}(t)k_{MTTP\_engine}, \tag{3}$$

where $P_{bat}(t)$ is the current charging and discharging power of PCS, $P_{wind}(t)$ is the current generating power of the fan, and $k_{MTTP\_wind}$ is the MTTP maximum power factor of the fan-connected inverter, and its value ranges from 0 to 100%. $P_{solar}(t)$ is the current power generation of PV, $k_{MTTP\_solar}$ is the MTTP maximum power factor of photovoltaic grid-connected inverter, and its value range is 0–100%. $P_{engine}(t)$ is the real-time power generation of the diesel gensets, and $k_{MTTP\_engine}$ is the maximum power factor of the diesel gensets with MTTP.

$\sum P_{load}(t)$ is the load value of the system at the current time, and the specific expression is as shown in Equation (4):

$$\sum P_{load}(t) = \sum P_{sensitive}(t) + \sum P_{non-sensitive}(t) + \sum P_{equipment}(t), \tag{4}$$

where $\sum P_{equipment}(t)$ is the power consumption of the equipment in the system at the moment, such as the inverter $\sum P_{sensitive}(t)$ and $\sum P_{non-sensitive}(t)$ are the sum of the sensitive loads and non-sensitive loads in the system at the current time.

For microgrid systems, the installed capacity is small, the permeability is high, which leads to the abnormal problems such as rapid diffusion. The frequency control strategy designed in this paper adopts the idea of partition control, which divides the frequency into several zones. Each zone adopts its own control strategy to achieve a frequency stability of 49.5–50.5 Hz. The specific frequency area is divided as shown in Table 3.

**Table 3.** Frequency division.

| Interval Name | Frequency Variation Range |
| --- | --- |
| CH | Above 52 Hz |
| BH | 50.5–52 Hz |
| AH | 50–50.5 Hz |
| AL | 49.5–50 Hz |
| BL | 48–49.5 Hz |
| CL | Below 48 Hz |

As shown in Table 3, when the frequency interval is within the range of AH or AL, the system does not need to intervene, and only needs to perform energy balance distribution to realize Equation (5). The ultimate goal is to achieve energy balance in the microgrid, that is, each the sum of the energy generated by the distributed generator is equal to the sum of the electric power of the load.

$$\sum P_{load}(t) = \sum P_{DG}(t), \ P_{BAT} = 0,. \tag{5}$$

### 3.1. Frequency Control Strategy for Area A

When the frequency is in the AH or AL area, it means that the frequency deviation within the permissible range of power quality, the system does not require excessive interference. However, for economical optimal performance considerations, the economic optimal performance inside the system should be maintained at this time. That is to reduce the use of diesel engine at the same time to maintain the battery charge and discharge state within a certain power, so that the battery charge level to maintain an optimal range to deal with possible frequency fault or voltage fault at any time.

### 3.2. Frequency Control Strategy for Area B

When the system is in the BH zone, indicates that the frequency slightly exceeds the permissible fluctuation range of the rated frequency. It is generally caused by excessive active power in the system, and this disturbance may be instantaneous. When it is detected that the system frequency enters the BH zone, it should not be operated immediately, which may cause the system to oscillate and reduce the robustness of the system. The control strategy is shown in Figure 2.

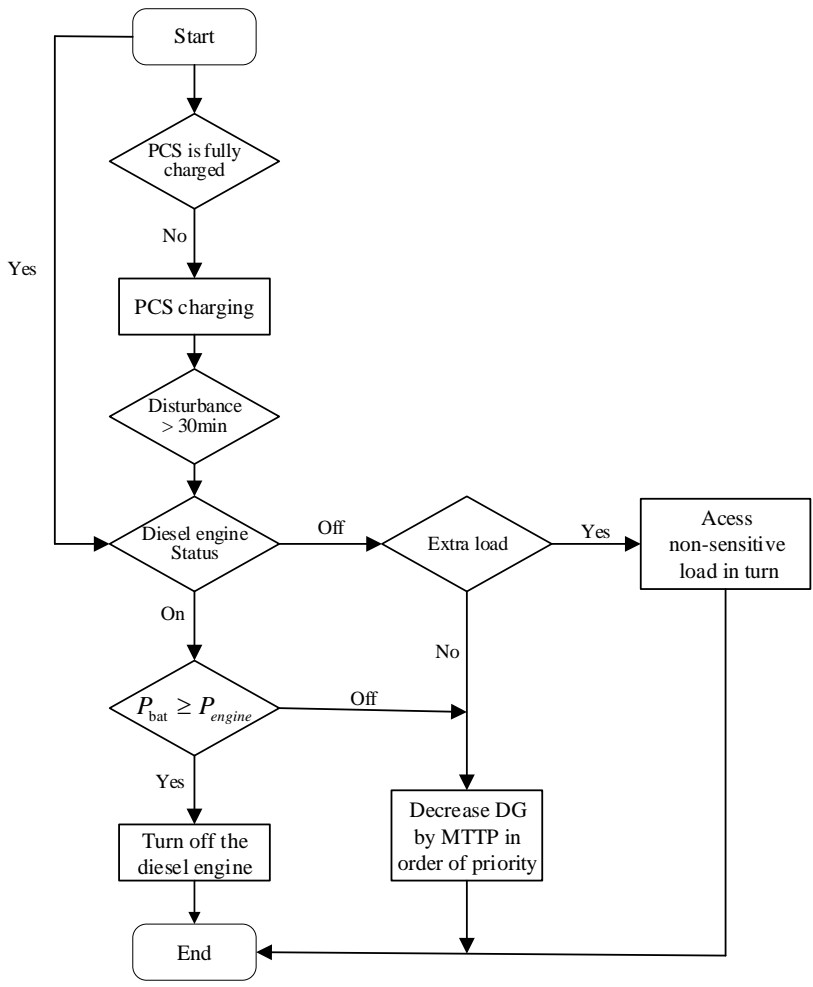

**Figure 2.** BH zone frequency control strategy.

When the system enters the BH zone, the operation is as follows:

(1)　Estimate the charge state of the PCS. If the PCS is in a chargeable state and is not full, go to step (2). If the PCS is already full and cannot be charged, then go to step (3).

(2)　Control the PCS for charging control, absorb excess active power, and jump to step (3).

(3)　If the disturbance lasts for more than 30 min, jump to step (4). Otherwise, end the control of the cycle and wait for the next cycle data.

(4)　If the diesel gensets is powered on at this moment, go to step (5), otherwise jump to step (6).

(5)　A simple evaluation of the power generated by the diesel gensets at this time. If the Equation (6) is satisfied, then jump to step (6), otherwise jump to step (8).

$$P_{BAT\_MIN} < P_{BAT}(t) + P_{engine}(t) < P_{BAT\_MAX}, \tag{6}$$

(6)　$P_{BAT\_MIN}$ is the charging power limit that PCS can withstand, and $P_{BAT\_MAX}$ is the discharge power limit that PCS can withstand.

(7)　Turn off the diesel gensets and reduce the PCS discharge power appropriately. End the control of the cycle and wait for the next cycle of data to make a judgment.

(8)　Decrease the power generation of the DG one by one according to the order of the DG priority. End the control of the cycle and wait for the next cycle of data to make a judgment.

(9)　If there is unloaded load in the system, go to step (9), otherwise jump to step (7).

(10)　Connect the load to the system one by one according to the order of priority of the load. End the control of the cycle and wait for the next cycle data to make a judgment.

When the frequency of the system is in the BL area, the control strategy is as shown in Figure 3.

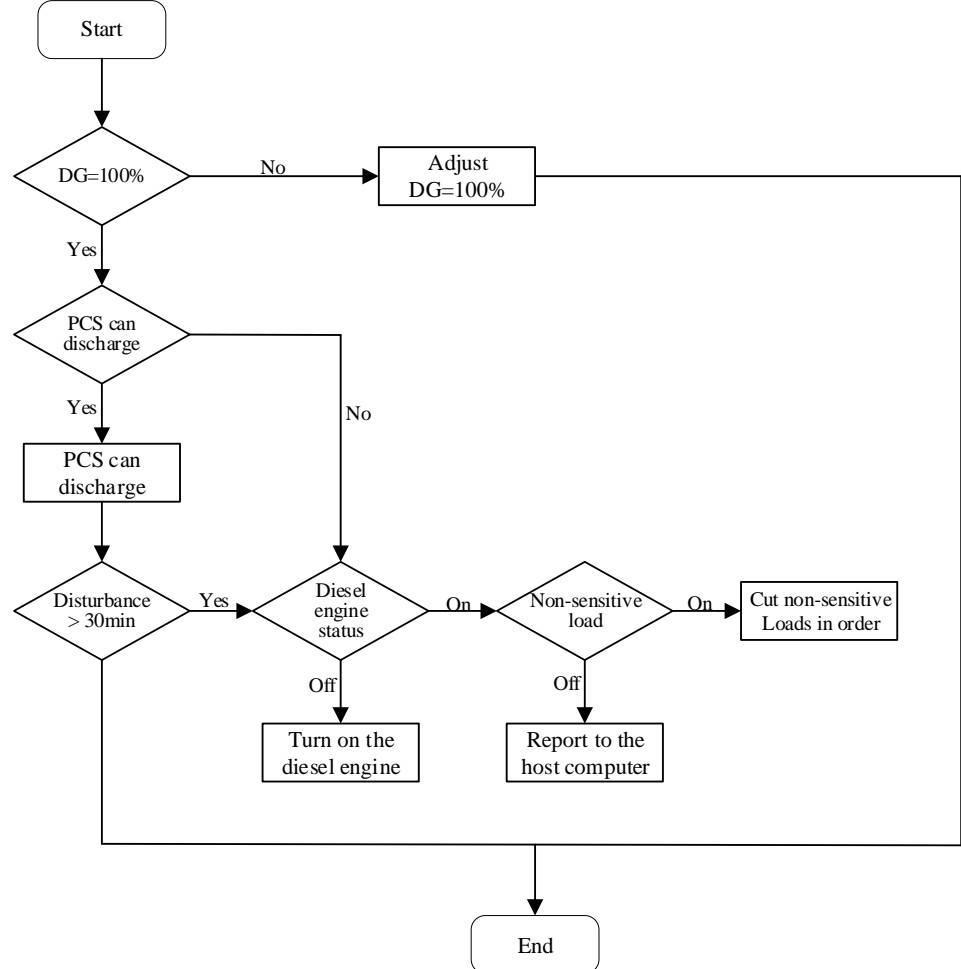

**Figure 3.** BL area frequency control strategy.

Among them, the specific steps are as follows:

(1) If the system frequency drops to the BL area, firstly check whether the distributed power supply in the system, such as photovoltaics, fans, etc., is 100% of the maximum output. If it is not 100% output, gradually increase its output coefficient to 100%. End the cycle control and wait for the next cycle data to judge, otherwise jump to step (2).

(2) When the BL zone disturbance occurs, the PCS state is judged. If the PCS does not reach the lower discharge limit, the process jumps to step (3), otherwise it jumps to step (4).

(3) Control the PCS to quickly release power, suppress system frequency fluctuations and jump to step (5).

(4) If the disturbance is less than 30 min, the PCS discharge state is maintained, end the cycle control and wait for the next cycle data to judge. Otherwise jump to step (5).

(5) If the diesel gensets is on, skip to step (6), otherwise turn on the diesel gensets, end the cycle control and wait for the next cycle data to judge.

(6) Judging the load condition in the system. If there is still non-sensitive load access, the non-sensitive load is cut off in order of power reduction. End the cycle control and wait for the next cycle data to judge. If the non-sensitive load is completely cut off, go to step (7).

(7) Report the upper-level monitoring system "system energy shortage in BL area", and maintain the output of all distributed generators in the system to maximize, waiting for the control command of the upper control system. End the cycle control and wait for the next cycle data to judge.

### 3.3. Frequency Control Strategy for Area C

When the system frequency is in the CH zone, it is a more dangerous state, and it is necessary to jump out of this zone in a short time to avoid the danger of system collapse. Therefore, it is necessary to evaluate the status of each DG and eliminate the fault load. The specific steps are shown in Figure 4. Among them, the specific steps are as follows:

(1) When the system detects that the frequency enters the CH zone, firstly check the status of the diesel gensets, and turn off the diesel generator if it's on. End the cycle control and wait for the next cycle data to judge, otherwise jump to step (2).

(2) Check each DG for failures one by one, and cut off DG if anything goes wrong. End the cycle control and wait for the next cycle data to make judge, otherwise jump to step (3).

(3) According to the descending power order of the generators, the DGs are cut one by one and reported to the upper-level monitoring system "System CH-zone failure", and jump to step (4).

(4) Check if the CH zone disturbance exceeds the specified time. If the specified time is not exceeded, maintain the system status and keep the system frequency as stable as possible. End the cycle control and wait for the next cycle data to judge. If the specified time is exceeded, skip to step (5).

(5) Shut down the microgrid system to protect critical equipment and generator equipment within the microgrid. Evaluate the power of the PCS and prepare for the black start plan. Report to the upper monitoring system "The system has a serious fault and has been shut down", waiting for the upper monitoring system black start command.

When the system frequency is in the CL area, it is a dangerous state and must jump out of this area in a short time. Otherwise, the system failure will cause damage to the internal equipment of the microgrid, triggering the island protection shutdown of DG and causing more serious consequences. Therefore, the control strategy in this interval is to quickly improve the internal power supply of the system, cut off all unnecessary loads, and quickly evaluate the distributed power failure state as the core idea. The specific process is shown in Figure 5.

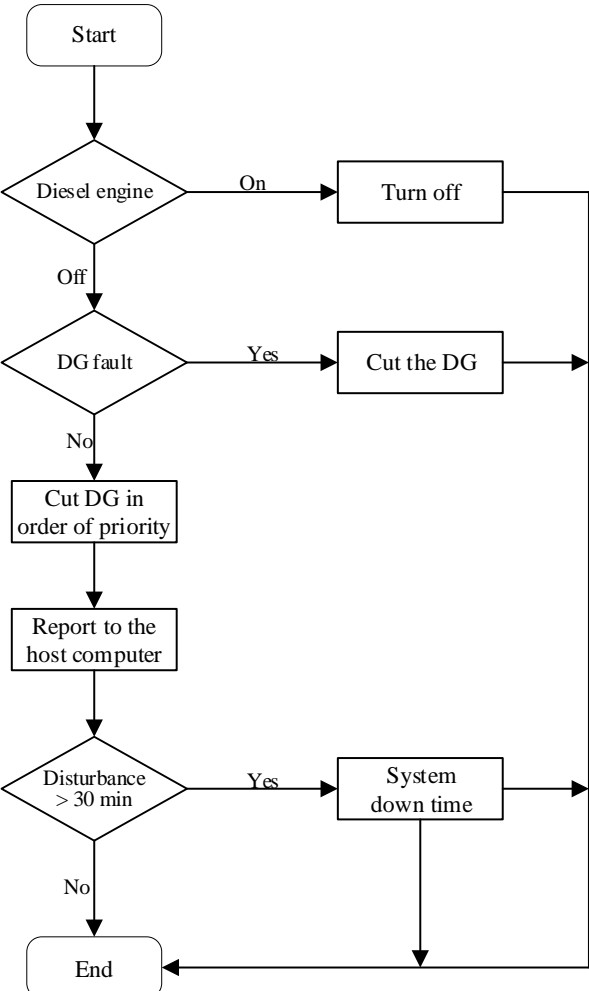

**Figure 4.** CH zone frequency control strategy.

The specific control steps are as follows:

(1) When the system detects of the system is in the CL area, it immediately performs discharge control on the PCS because the PCS has a fast transient response performance and jump to step (2).

(2) Detect the state of the diesel genset. If it is off, turn on the diesel genset to supply power to the system, end the control cycle and wait for the next cycle of data to judge. If the diesel genset is on, go to step (3).

(3) Check if other DGs are in 100% output state, if they are not in 100% output state, adjust their MTTP coefficient to 100% immediately, end the control cycle, and wait for the next cycle data to judge. If it is in the 100% output state, go to step (4).

(4) If there is a non-sensitive load currently connected to the microgrid system, immediately remove all non-sensitive loads, end the control cycle, and wait for the next cycle of data for judgement. If no non-sensitive load is connected to the system, jump to step (5).

(5) Check whether the system DG is faulty, whether there is a fault in the primary load and take the operation of cutting out the system for the corresponding faulty equipment then jump to step (6).

(6) Estimate the stability of the system. If the Equations (7) and (8) are satisfied, the frequency of the system is still in the offset state and the situation is deteriorated.

$$\Delta f(n) = f(t) - f(t-1), \tag{7}$$

$$\sum_{n-3}^{n} f(n) < 0. \tag{8}$$

It indicates that the frequency of the system is still in the offset state and the situation is deteriorating. At this point, it should be transferred to step (7), and report "system frequency CL area fault" to the upper monitoring system, try to maintain the stable state of the system, and wait for the command of the upper monitoring system. End the cycle operation and wait for the next cycle data to make a decision.

(7) If the system frequency is in a state of continuous deterioration, the microgrid system is shut down in order to avoid equipment damage inside the system. Reported to the host computer "The system has been stopped, in the CL area frequency failure", while assessing the PCS status, do a black start plan, waiting for the upper monitoring system boot command.

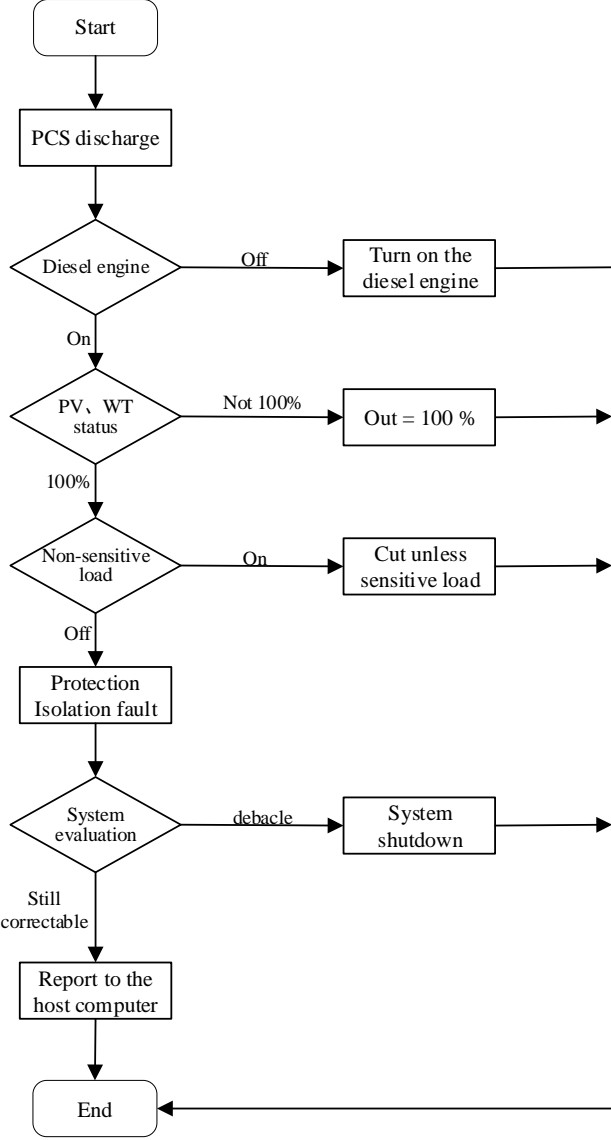

**Figure 5.** CL zone frequency control strategy.

## 4. Verification Analysis

In order to verify the proposed partition frequency stability control strategy in this paper, the stable operation test was carried out in microgrid laboratory of Shandong University (Weihai). The test was

conducted in the Weihai campus of Shandong University in Weihai city (37°31′51.1′′ N 122°03′38.7′′ E), Shandong Province, China. The test time is 7 March 2019. The test period is from 9:00 am to 15:00 pm. The test duration is 6 h and the sampling period is 50 s. On the day of the test, the weather was 6–7 north wind, and the temperature was 1–2 °C. The cloudy weather turned to clear with slight haze.

For the stable operation test of the islanded microgrid in this paper, firstly, PCS is started to provide reference voltage and reference frequency for the remaining distributed generators. Input to primary load and photovoltaic generator set; after the system is relatively stable, input non-sensitive load. After the grid-connected voltage of the wind machine reaches the requirement, the wind turbine is started to be connected to the grid for power generation.

In this test phase, the system bus frequency changes as shown in Figure 6. The power output of the photovoltaic generator set, wind turbine, and energy storage device is shown in Figures 7–9.

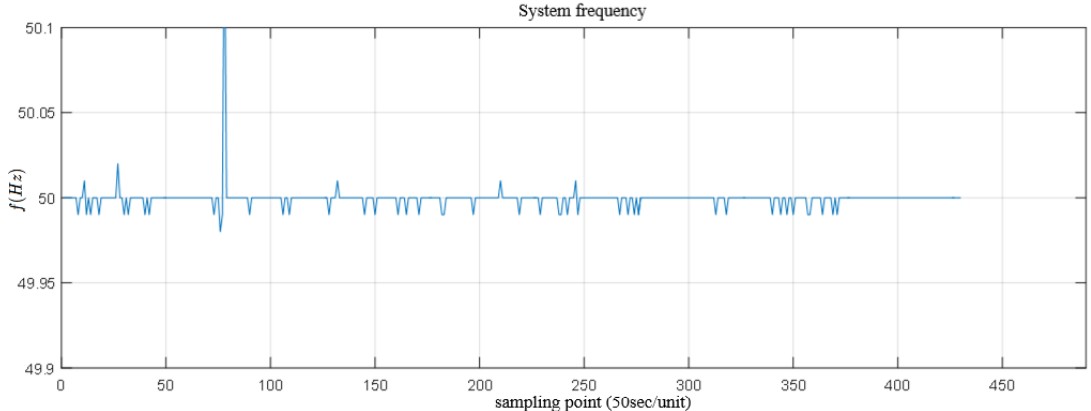

**Figure 6.** Frequency variation during the energy balance test phase.

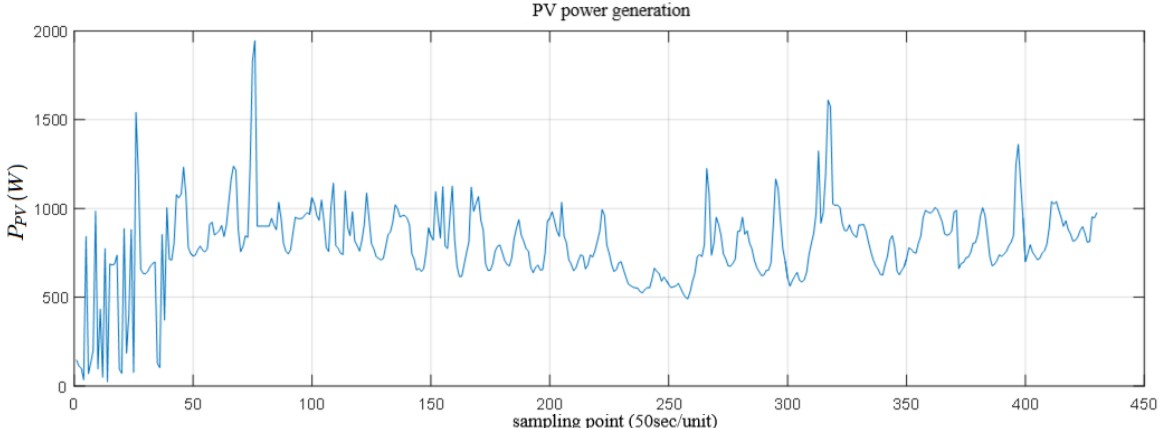

**Figure 7.** Photovoltaic power and energy deployment.

As shown in Figure 6, during the test period, the system frequency fluctuates around 50 Hz and is in a relatively stable state. The energy in the test is in the balance stage, and the power frequency conforms to the power consumption standard. By comparison with the variation of photovoltaic power generation in Figure 7, it can be seen that near the 80th sampling point, the photovoltaic power is relatively high, producing an obvious energy impact on the system frequency and causing a peak in the system frequency. In combination with the change of battery charging and discharging power in Figure 9, when the system frequency peak is generated near the 80th sampling point, the battery will charge quickly to absorb excess electric energy. When the power of photovoltaic power generation falls off the cliff, the battery can quickly discharge and replenish the system's electric energy, so that the power frequency of the system can quickly restore stability.

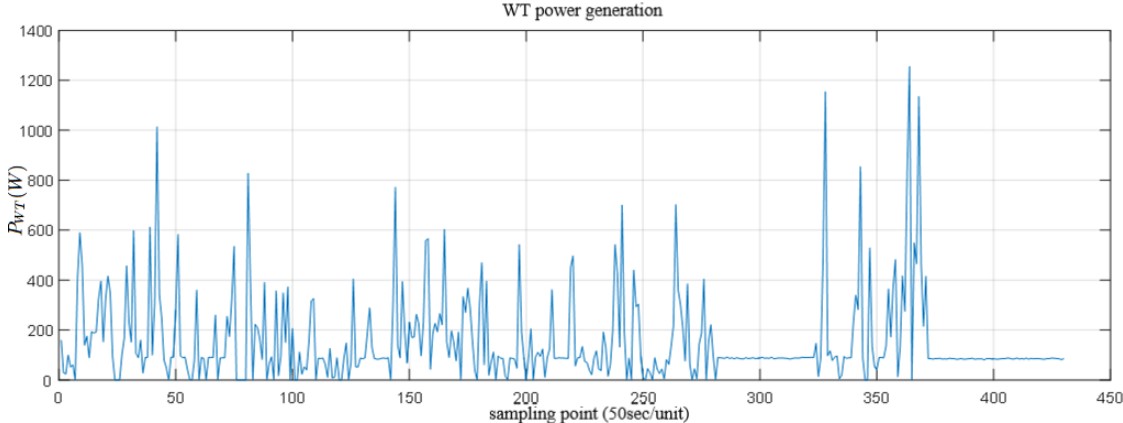

**Figure 8.** Wind turbine generator power and energy deployment.

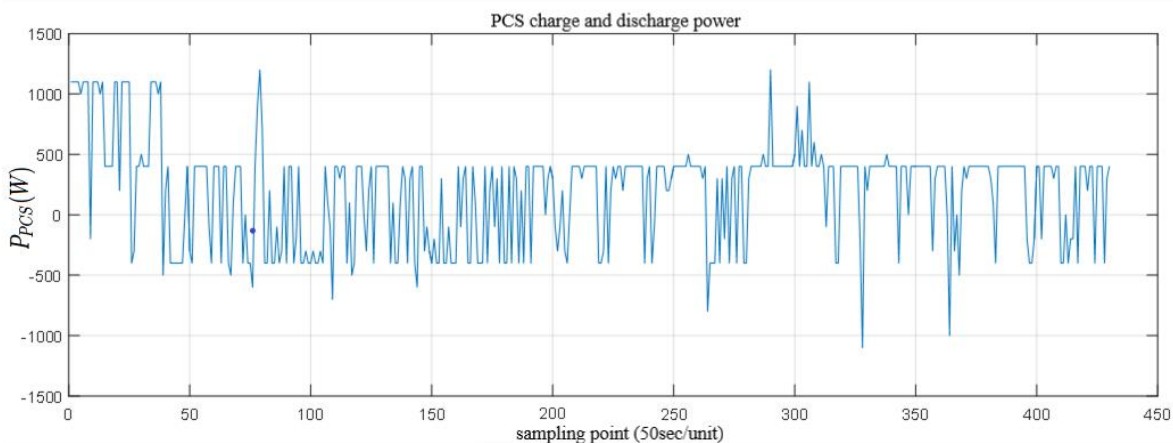

**Figure 9.** Schematic diagram of power conversion systems (PCS) charge and discharge power changes.

Figure 7 shows the variation of photovoltaic power generation. Since the weather on the day of the test was cloudy and turned sunny, it can be seen from Figure 6 that under cloudy weather conditions, the output power of photovoltaic power generation presents a certain randomness and variability, which may constitute a certain energy impact on the system.

As shown in Figure 8, the variation of the generating power of the wind turbine shows that the short-term periodicity of the output power characteristic of wind power generation is not obvious, showing greater randomness and variability than photovoltaic power generation, and the generating capacity is not stable. By contrast with the variation of system bus frequency in Figure 6, it can be seen that in this test experiment, wind power generated multiple disturbances to the system frequency, but the disturbance amplitude to the system frequency was small due to the small power generated.

Figure 9 shows the change of battery charging and discharging power. In the figure, when the battery power is less than 0 W, it is in the charging state; when the power is greater than 0 W, it is in the discharging state. Compared with Figures 7 and 8, when the total power of solar power generation and wind power generation is greater than the supply demand, the battery is in the charging state to absorb excess electric energy; on the contrary, the battery is in the discharge state to supplement the supply demand, and the response speed is fast. Since the strong randomness of the output power of photovoltaic power generation and wind power generation. In order to smooth out these large energy fluctuations quickly, the battery may exceed the range of ±500 W in a short time. In an overall view, the island-type microgrid through the central controller to properly adjust the load and energy, effectively control the battery's charge and discharge power in the power range of −500 W (charge)

to 500 W (discharge), effectively ensuring the islanded microgrid bus while the frequency is stable, the charge and discharge pressure on the battery is also reduced.

## 5. Conclusions

This paper proposes a frequency cutter de-loading control strategy based on the partition control idea for the islanded AC microgrid containing high-power diesel gensets. The diesel gensets can support the frequency stability in the system, and it will not cause the system large impact, which affects the power quality of the system. At the same time, the priority of distributed power and load is considered in the frequency control strategy, which can realize more economical dispatching and generation tasks, protect the use of important loads, and complete system power distribution and balance. The experimental results show that the proposed method increases the flexibility and stability of the microgrid system, reduces the operating cost and contributes to the development of islanded microgrid in the future.

The frequency control strategy proposed in this paper aims to provide high quality power supply for remote mountainous areas, small islands and other areas that cannot be covered by traditional large power grid. In the future, through actual visits and surveys of remote island areas around Weihai, based on this control strategy, an island capacity matching scheme based on the electricity demand of fishery and aquaculture can be determined. In addition, as an important part of the frequency control of the islanded microgrid, the service life of the battery is related to the economic benefits of the islanded microgrid. In the future, the use optimization strategy of the battery should be further studied. Furthermore, the proposed control strategy should be modified so that it can be applied to the power management of grid-connected microgrids and broaden the application field.

**Author Contributions:** Y.L. and S.W. proposed innovative idea; Y.L. conceived the algorithm and wrote the first draft; X.W. and Y.L. improved the algorithm; X.W. performed the experiments; Y.L. and S.W. analyzed the results; Y.L. and X.L. drafted the manuscript; X.W. provided writing advice; S.W. and X.W. All authors have read and agreed to the published version of the manuscript.

**Funding:** The research received no external funding.

**Acknowledgments:** We gratefully acknowledge the technical assistance of DL850E ScopeCorder.

**Conflicts of Interest:** The authors declare no conflict of interest.

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
