# Peer review of "Research on Frequency Control of Islanded Microgrid with Multiple Distributed Power Sources"

_processes, doi:10.3390/pr8020193_

Round 1
Reviewer 1 Report
The manuscript entitled „Research on Frequency Control of Islanded 2 Microgrid with Multiple Distributed Power Sources” presents up-to-date and important topic. The manuscript is well written and present clear and sufficient literature review. The experiment are designed and performed very accurate and transparently outlined. However I have two comments. Firstly, in lines 42, 47, 49 the word “Literature” should be replaced by i.e. “Literature studies showed that”. Moreover, Figures 6-9 should be improved because axis named are not visible.
In my opinion the paper entitled „ Research on Frequency Control of Islanded 2 Microgrid with Multiple Distributed Power Sources” is interesting and well written but has to be improved . Overall, I suggest publication of this paper in PROCESSES after minor revision.
Author Response
Reply to Reviewer’s Comments
Reply to Reviewer #1
Firstly, we would like to thanks the reviewer for the positive and constructive comments. According to your comments, we have checked our manuscript carefully. Some grammatical & language errors and other inexact expressions in the manuscript have been corrected. The important changes in our revised manuscript have been marked “in Red”. Thanks again.
Reviewer’s Comments:
Point 1: In lines 42, 47, 49 the word “Literature” should be replaced by i.e. “Literature studies showed that”.
Response 1: Thanks for your professional advice, according to your suggestion we modified the corresponding parts of the manuscript and marked them in red.
Point 2: Figures 6-9 should be improved because axis named are not visible.
Response 2: Thank you for your careful review, according to your suggestion we change the horizontal axis label of figures 6-9 to "sampling point(50sec/unit)" which could clearly reflect the sampling method of sampling points. At the same time, we modified the vertical axis label to be an abbreviation of its specific content, so as to clearly express its meaning.
At last, special thanks to you for your comments again!

Reviewer 2 Report
This looks a useful paper, proposing a decision system for which power sources to use, depending on which band the frequency is in. It also reports on a test of the method on a microgrid at Shandong University.
I am not sufficiently knowledgeable in the field to comment on the novelty of their research, but it sounds good to me.
One minor question is that in Figure 9, the power into and out of the battery store is mostly constrained to lie between plus and minus 500W, but there are periods during which it leaves this band. What is the operating principle for the battery store? It is also strange to me that the battery spends no time at zero power. Perhaps it is a large one, but for optimal use of a battery I would expect substantial periods when it is fully charged or fully discharged. Can the authors comment on this?
After responses to these two questions I would recommend publication.

Author Response
Reply to Reviewer’s Comments
Reply to Reviewer #2
Firstly, we would like to thanks the reviewer for the positive and constructive comments. According to your comments, we have checked our manuscript carefully. Some grammatical & language errors and other inexact expressions in the manuscript have been corrected. The important changes in our revised manuscript have been marked “in Red”. Thanks again.
Reviewer’s Comments:
Point 1: In Figure 9, the power into and out of the battery store is mostly constrained to lie between plus and minus 500W, but there are periods during which it leaves this band. What is the operating principle for the battery store?
Response 1: Thanks for the comments. The operating principle of the battery store is that, when the frequency of the Microgrid system is in a relatively stable range (area A or B), the load balance in the system is controlled by the battery charge or discharge in preference. This is because the battery reaction speed is the fastest, and the charge-discharge power is small, which could quickly stabilize the small frequency fluctuation of the microgrid system.
Keeping the power of the battery within the range of ±500W could reduce the pressure of charging and discharging of the battery. However, due to the strong randomness of the output power of photovoltaic power generation and wind power generation, it may cause a large energy shock or energy trough to the system in a short time. In order to smooth out these large energy fluctuations quickly, the battery may exceed the range of ±500W in a short time. We have made some supplementary notes in the lines 305~308 of the manuscript. Thanks for your suggestions again.
Point 2: The battery spends no time at zero power. Perhaps it is a large one, but for optimal use of a battery I would expect substantial periods when it is fully charged or fully discharged.
Response 2: Thanks for the comments and valuable suggestions. As mentioned in the previous problem, because of the quick response time of the battery, it is preferred to control the internal frequency balance of the microgrid. However, since photovoltaic power generation and wind power generation have strong randomness and intermittency, it is difficult to achieve Eq. (2) (line110) only by photovoltaic power generation and wind power generation. Therefore, the battery is frequently in the state of low power charge or discharge, so the battery spends no time at zero power.
In addition, as you said, the regular battery for the "fully charged or fully discharged " operation can reduce the battery internal polarization and extend the battery’s life span. The optimal control strategy of battery is also a focus of current research in the field of islanded microgrid. However, this paper focuses on the frequency control strategy of microgrid, so there is no explanation about the optimal control of battery. We will carry out research on the optimization of battery in the future work.
At last, special thanks to you for your comments again!
